# Whole-Genome Investigation of *Salmonella* Dublin Considering Mountain Pastures as Reservoirs in Southern Bavaria, Germany

**DOI:** 10.3390/microorganisms10050885

**Published:** 2022-04-23

**Authors:** Corinna Klose, Nelly Scuda, Tobias Ziegler, David Eisenberger, Matthias Hanczaruk, Julia M. Riehm

**Affiliations:** 1Bavarian Health and Food Safety Authority, Eggenreuther Weg 43, 91058 Erlangen, Germany; corinna.klose@lgl.bayern.de (C.K.); nelly.scuda@lgl.bayern.de (N.S.); tobias.ziegler@lgl.bayern.de (T.Z.); david.eisenberger@lgl.bayern.de (D.E.); 2Bavarian Health and Food Safety Authority, Veterinaerstrasse 2, 85764 Oberschleißheim, Germany; matthias.hanczaruk@lgl.bayern.de

**Keywords:** *Salmonella* Dublin, mountain pasture, whole-genome sequencing, genome analysis, antimicrobial resistance, virulence, cattle

## Abstract

Worldwide, *Salmonella* Dublin (*S.* Dublin) is responsible for clinical disease in cattle and also in humans. In Southern Bavaria, Germany, the serovar was identified as a causative agent for 54 animal disease outbreaks in herds between 2017 and 2021. Most of these emerged from cattle herds (*n* = 50). Two occurred in pig farms and two in bovine herds other than cattle. Genomic analysis of 88 *S.* Dublin strains isolated during these animal disease outbreaks revealed 7 clusters with 3 different MLST-based sequence types and 16 subordinate cgMLST-based complex types. Antimicrobial susceptibility investigation revealed one resistant and three intermediate strains. Furthermore, only a few genes coding for bacterial virulence were found among the isolates. Genome analysis enables pathogen identification and antimicrobial susceptibility, serotyping, phylogeny, and follow-up traceback analysis. Mountain pastures turned out to be the most likely locations for transmission between cattle of different herd origins, as indicated by epidemiological data and genomic traceback analyses. In this context, *S.* Dublin shedding was also detected in asymptomatic herding dogs. Due to the high prevalence of *S.* Dublin in Upper Bavaria over the years, we suggest referring to this administrative region as “endemic”. Consequently, cattle should be screened for salmonellosis before and after mountain pasturing.

## 1. Introduction

More than 2600 serovars are known in the genus of *Salmonella* [1]. Most of them are pathogenic for humans and animals belonging to the zoological classes of mammals, birds, and reptiles [2,3,4]. Animals are known to represent reservoirs for these Enterobacteriaceae; however, environmental or inanimate vehicles can also be part of the transmission chain [5,6,7]. Strong characteristics regarding *Salmonella* infection and the respective clinical picture for the serovar Dublin are still carrying and subclinical shedding [8,9,10]. The carrier state may persist for several years. Intermittently triggered shedding may lead to broad spreading of the pathogen on pastures [9]. When taking into account that *Salmonella* can survive in nonhost environments for months, infection of herbivores grazing on pastures is likely, thus possibly completing a transmission cycle [9,11]. According to one study published in Austria, cattle probably acquired salmonellosis by grazing on contaminated alpine pastures [12]. The *Salmonella enterica* subspecies *enterica* serovar Dublin (*S.* Dublin) is primarily adapted to cattle. The most frequently described clinical manifestations are abortions in cows, whereas acute systemic disease and diarrhea are predominant in calves [9,13].

In the rare case of zoonosis, however, *S.* Dublin can cause invasive infection with high mortality rates [14,15,16,17]. As published, *S.* Dublin is the most common serotype that causes systemic salmonellosis in humans (61%) in the United States, and the proportion of resistant isolates is higher than among other serotypes [18,19]. In addition to antimicrobial resistance, *S.* Dublin may harbor virulence factors that contribute to the pathogenicity of the respective isolates [20]. Globally, this pathogen is considered a public health threat [18,21]. In Germany, *S.* Dublin caused 30–45% of all salmonellosis outbreaks in cattle herds in the years 2017–2019 [22,23]. Federal public health departments reported 25 cases of human infection with *S.* Dublin in Germany during the period of our study from 2017 to 2021, of which four cases occurred in Bavaria [24].

As a consequence, and in line with One Health, *Salmonella* outbreaks among humans and animals as well as incidences regarding food and feed are strictly monitored in Europe. In Germany, legislated *Salmonella* control in cattle was initiated in the 1970s [25]. In all of Europe, the registration and subsequent assignment of a unique identification number to each individual cow is mandatory to ensure traceback options. If *Salmonella* is detected in cattle, local veterinary authorities have to place the herds under restriction, initiate remedial measures, and conduct an epidemiological investigation. The source of infection, however, often remains unknown [26]. Mountain pastures offer a challenge in epidemiological investigations. For a limited time during the summer months and the associated geographic relocation of animals to mountain pastures, the ownership of cattle is temporarily transferred to the proprietor of the pasture. Therefore, several owners must be considered. Investigations including whole-genome analysis may provide valuable clues as to transmission routes in these circumstances.

## 2. Materials and Methods

### 2.1. Salmonella Isolates

In the present study, the definition of a *Salmonella* animal disease outbreak was the detection of the bacterium in one or more animals on a specific farm during the following calendar year. For the special situation of a mountain pasture, the demarcation of an outbreak was defined by the designated pasture grounds, where animals from different farms stayed in close contact during the summer months.

As regulated by German law, all Bavarian cattle farms associated with *Salmonella* outbreaks were exclusively investigated cost-free at the federal state veterinary laboratories of the Bavarian Health and Food Safety Authority to ensure transparency. In the present study, sample types included feces, fecal swabs or tissue from cattle, bovines other than cattle, pigs, and herding dogs in Southern Bavaria. The diagnostic algorithm for *Salmonella* was applied according to standardized methods, including the ISO Standard 6579-1:2020-08 [27]. Suspicious colonies were identified using MALDI-TOF mass spectrometry (Bruker, Bremen, Germany). All *Salmonella* isolates were serotyped pursuant to the White–Kauffmann–Le Minor scheme by using poly- and monovalent anti-O and anti-H sera (Sifin, Berlin, Germany) [28]. The isolates were preserved at −80 °C by the Bavarian Health and Food Safety Authority.

In the 5-year period of 2017–2021, *S.* Dublin index isolates originating from 54 individual animal disease outbreaks were included in the study. In a further approach, 3 isolates recovered from a mountain pasture outbreak at Miesbach in 2020 and 28 isolates obtained from a single outbreak in the summer of 2021 on a mountain pasture in Garmisch-Partenkirchen were investigated more profoundly. To address other questions, such as transmission between different animal species on farms or the genomic changes of the bacterium in hosts over time, additional isolates collected from the animal disease outbreaks were included. A total of 88 *S.* Dublin isolates were investigated in the present study.

Two of the investigated isolates (54 and 57) have previously been analyzed, and the results were published in a nationwide study by the Institute of Bacterial Infections and Zoonoses, Friedrich-Loeffler-Institute, Jena, Germany [29].

### 2.2. Epidemiological Analysis

Subsequent to the isolation of *Salmonella*, additional contact animals were investigated during an epidemiological analysis. Data on herd identity, ownership, cattle trade, date of death or slaughter, and geographical location of the individual animals were obtained from the national Animal Traceability Information Management System [30] and used for the epidemiological analysis in the present study. For the epidemiological analysis in the present study, the previous owner, the pasture proprietor, and the owner of the specific animal after the pasturing period were determined. Finally, upon considering all involved geographic locations, all data were compiled up to the *Salmonella* isolation date. The clustering of isolates based on whole-genome analysis was matched to owners, geographic locations, and respective years to identify individual animals or geographic locations as possible outbreak origins. The federal Animal Disease Information System [31] consistently collects data on all contagious animal diseases, mandatorily reported by the local authorities, as a direct information platform for the federal Ministry of Food and Agriculture. This source was used to verify the results obtained in the epidemiological analysis.

### 2.3. Antimicrobial Susceptibility Testing

In vitro antimicrobial susceptibility testing was conducted on 71 isolates, whereby 1 isolate of each animal host species per outbreak was included. In the Garmisch-Partenkirchen mountain pasture instance, 1 isolate per originating farm and cluster was investigated. Nine antimicrobial substances from six classes were tested (Figure 1) using the microbroth dilution method in accordance with the manufacturer’s instructions (Micronaut-S MDR MRGN-Screening and Micronaut-S Grosstiere 4, Merlin, Bornheim, Germany). The minimum inhibitory concentration was determined using a photometric plate reader system (Tecan, Männedorf, Switzerland) and MCN6 software (Sifin, Bruker, Bornheim, Germany) pursuant to the protocols of the Clinical and Laboratory Standards Institute Vet01S, 5th ed. (CLSI, Wayne, PA, USA) [32]. *Escherichia coli* ATCC 25922 was used as a reference strain for quality control purposes. Antimicrobial susceptibility results were interpreted according to CLSI breakpoints of numerical MIC values, which were established for specific host–pathogen–drug combinations. In application of these breakpoints, the results were categorized as susceptible, intermediate, or resistant phenotypes for each specific substance.

### 2.4. Whole-Genome Sequencing

To implement whole-genome sequencing, a single colony of an overnight blood agar plate culture at 37 °C was used for genomic DNA extraction, applying a DNeasy Blood and Tissue Kit (Qiagen, Hilden, Germany) in accordance with the manufacturer’s instructions. Purified DNA was quantified using a Qubit 4.0 fluorometer (Invitrogen, Carlsbad, CA, USA) via a Qubit dsDNA HS Assay Kit (Life Technologies, Waltham, MA, USA). Library preparation was performed using a Nextera XT DNA Library Prep Kit and Nextera XT Index Kit (Illumina, San Diego, CA, USA). Libraries were quantified using an Agilent High Sensitivity DNA Kit (Agilent Technologies, Waldbronn, Germany) on a 2100 Bioanalyzer Instrument (Agilent Technologies, Santa Clara, CA, USA). Subsequently, 2 × 150 bp paired-end reads were generated on an Illumina MiniSeq system, and sequenced reads with a mean assembled coverage depth of 98× (range 37–176×) were analyzed.

### 2.5. Genetic Characterization

Verification of the *Salmonella enterica* subspecies *enterica* and in silico serotype prediction were carried out using Mash distance implemented in SeqSphere + software version 7.0.2 (Ridom GmbH, Münster, Germany) and the SeqSero 1.2 online tool [33,34]. Furthermore, screening of genetic elements coding for antimicrobial resistance was performed by applying the publicly available database ResFinder 4.1 with an ID threshold of 90% and a minimum length of 60% [35]. Moreover, the virulence factor database (VFDB) was used to screen for putative virulence factors with an ID threshold of 85% and a minimum length of 60% [36]. Plasmid replicons were identified by PlasmidFinder 2.1 with an ID threshold of 95% and a minimum length of 60% [37].

Core genome multilocus sequence typing (cgMLST) was based on a gene-by-gene allele calling of 3002 target genes with quality parameters as previously established for *Salmonella* isolates [38]. The in silico MLST was conducted using SeqSphere+ software. To visualize the clonal relationship between isolates, a minimum spanning tree was created. For this, the “pairwise ignoring missing values” option was turned on when using the abovementioned software. Accordingly, closely related isolates with a maximum difference of seven alleles were subsequently assigned to clusters.

To investigate the epidemiology of the mountain pasture outbreak in Garmisch-Partenkirchen and to zoom into the molecular background of the involved isolates, a single-nucleotide polymorphism (SNP)-based phylogenetic analysis of Cluster 1 was performed using BioNumerics software version 7.6 (Applied Maths NV, Sint-Martens-Latem, Belgium). Initially, sequencing reads of Isolate 1 of the investigated panel were used as the index and mapped to the reference strain *Salmonella enterica* subspecies *enterica* serovar Dublin strain ATCC 39184 (NCBI accession CP019179.1). Thereafter, the sequencing reads of all Cluster 1 isolates were mapped to the index isolate using the default settings. Subsequently, SNPs were determined using the following SNP filtering settings: (i) each retained SNP position has a minimum of 10 × coverage, (ii) the minimum distance between retained SNP positions is 12 bp, and (iii) nondiscriminatory positions between the isolates are removed. Retained SNPs served as a basis for phylogenetic cluster analyses to determine the SNP similarity matrix.

The raw sequencing data were deposited at the European Nucleotide Archive (ENA project number: PRJEB50766).

## 3. Results

### 3.1. Identification and Characterization of S. Dublin

A total of 54 individual animal disease outbreaks caused by *S.* Dublin were detected in Southern Bavaria between 2017 and 2021. Outbreaks occurred on 48 cattle farms, 2 pig farms, and 2 farms keeping bovines other than cattle (Table 1 and Appendix A). One outbreak occurred among cattle of different originating farms on a mountain pasture in the district of Miesbach (mp-MB) in 2020. Another occurred on a mountain pasture in the district of Garmisch-Partenkirchen (mp-GAP) in 2021. Due to issues regarding transmission, the 2 outbreaks on mountain pastures were investigated more closely and including all recovered isolates in this study. A total of 88 *Salmonella* isolates were processed, and all were identified as belonging to the Dublin serotype using in vitro and in silico methods (Appendix A).

### 3.2. Antimicrobial Susceptibility

For a subset of 71 isolates, the in vitro antimicrobial susceptibility testing yielded susceptible growth regarding the substances gentamicin, amoxicillin–clavulanate, ampicillin, meropenem, ciprofloxacin, levofloxacin, and trimethoprim–sulfamethoxazole. In 3 isolates, 7, 19, and 20, the minimum inhibitory concentration (MIC) revealed an intermediate value for chloramphenicol, whereas all other isolates were susceptible. For Isolate 8, a resistant phenotype was determined for tetracycline. This was the only isolate that showed antimicrobial resistance in this study (Figure 1, Appendix A). Regarding the in silico analysis, all isolates revealed the *aac (6′)-Iaa* gene, which is a silent, chromosomally encoded aminoglycoside acetyltransferase. No further matches in the screened genes coding for antimicrobial resistance were detected using the ResFinder 4.1 database (Appendix A).

### 3.3. In Silico Analysis for Virulence and Plasmid Content

The molecular screening for 130 putative virulence factors in silico revealed Gifsy-2 prophage, which encodes a superoxide dismutase (*sodCI*), in all isolates. The chromosomal locus *viaB*, coding for genes for Vi antigen, a capsular polysaccharide virulence antigen, was detected only in all isolates belonging to MLST sequence type (ST) 1494 (*n* = 10). All of the 88 investigated isolates harbored the plasmids IncX1 and IncFII (S), which are associated with virulence (Appendix A). The IncA/C2 plasmid associated with multidrug resistance was not found in any of the isolates.

### 3.4. Genomic and Epidemiological Analysis of Animal Disease Outbreaks

This study investigated 54 *S*. Dublin outbreaks that occurred on individual farms and 2 that occurred on mountain pastures in Southern Bavaria from 2017 to 2021 (Table 1 and Appendix A). All animal disease outbreaks occurred in Upper Bavaria with the exception of a farm keeping solely pigs in Lower Bavaria (Figure 2).

Genotyping was used to characterize all *S*. Dublin strains. The 88 isolates were assigned to three known MLST STs: 10, 1487, and 1494. Furthermore, they were assigned to 16 subordinated *S.* Dublin cgMLST complex types (CTs): 1322, 1324, 6738, 6742, 6745, 8614, 8615, 8616, 8620, 8623, 8626, 8719, 9033, 9034, 9043, and 9163 (Table 1 and Appendix A, Figure 3). The genotyped isolates were grouped into seven clusters. Three single isolates could not be assigned to any of these clusters (Table 1 and Appendix A).

**Cluster 1** contained 52 isolates originating from all over Southern Bavaria. The sampling dates covered the entire 2017–2021 investigation period. Cluster 1 consisted of ST 10 and 6 different CTs: 1322, 6745, 8614, 8719, 9034, and 9163 (Table 1, Figure 3 and Figure 4). Furthermore, SNP typing was carried out to trace back the isolates within this cluster (Figure 5, Appendix A).

All isolates from the mountain pasture outbreak in the district of Garmisch-Partenkirchen, mp-GAP, formed unique SNP branching within Cluster 1 and revealed a genetic distance of 2.9 different SNPs at most (Figure 5, Appendix A). This outbreak involved 22 cattle from 6 different farms and 2 herding dogs. The sampling yielded 28 ST 10 isolates. These were assigned to Clusters 1 and 2 and revealed the 3 different CTs 6742, 6745, and 9163. One dog was sampled twice, which revealed genetically identical isolates (38 and 83), whereby the latter was collected 4 weeks later. Duplicate samples from 2 cows taken 2 weeks apart also revealed identical genotypes (39 and 71; 75 and 82). Isolates 40 and 73, however, were collected from the same cow 2 weeks apart and revealed the different CTs, 6745 and 9163 (Table 1 and Appendix A).

Regarding Farm 49 in the district of Bad Tölz-Wolfratshausen, 4 isolates within Cluster 1 were investigated in the present study (Appendix A). Three isolates (87, 88, 89) from 2021 clustered in the distinct mp-GAP group and revealed a ST 10/ CT 6745 genotype. The fourth isolate, Isolate 56, originated from 2017 and belonged to a different CT 1322 (Appendix A).

Epidemiological data revealed that animals of Farm 56 in the district of Garmisch-Partenkirchen were kept on the mp-GAP every summer. One of the 2021 *S.* Dublin-positive cows stayed on the mp-GAP in 2015, 2016, 2017, and 2021. Two isolates (52 and 86) originated from this farm and were collected in 2017 and 2021, and the same genotype ST 10/CT 6745 was identified (Appendix A). Isolate 36 revealed a similar genotype as other strains from the mp-GAP origin. However, the epidemiological analysis did not prove any contact between involved animals.

Isolates 2 (cattle) and 4 (herding dog) from one farm in the district of Garmisch-Partenkirchen revealed the same genotype ST 10/CT 1322, originating from a 2019 outbreak. These isolates clustered with Isolate 48 from 2017 and even revealed the same SNP allele profile (Figure 4 and Figure 5). The epidemiological analysis could not ascertain a possible transmission route due to the time discrepancy and missing animal connection.

In 2020, one animal disease outbreak occurred on a farm solely keeping pigs in the district of Landshut. This was the only outbreak to originate in Lower Bavaria. The respective genotype of Isolate 22 matched ST 10/CT 6745, which constitutes the most frequently found genotype in the study (Table 1). Two more isolates (19 and 24) revealed an identical SNP allele profile (Figure 5). These two originated from different farms, both in the district of Bad Tölz-Wolfratshausen (Appendix A). All three strains were isolated in 2020. No further links were found in the epidemiological analysis.

Isolates 46 and 47 originated from farms in the district of Garmisch-Partenkirchen in 2017. The genotypes of these isolates were identical ST 10/CT 1322 (Appendix A).

Isolate 43 clustered with Isolate 44, originating from the districts of Bad Tölz-Wolfratshausen and Mühldorf am Inn. Both isolates were collected in 2017 and matched to the ST 10/CT 6745 genotype, but no further epidemiological correlation was identified (Appendix A).

**Cluster 2** consisted of 17 isolates originating from 16 farms in the districts of Rosenheim, Miesbach, and Garmisch-Partenkirchen, all in Upper Bavaria. All isolates exclusively revealed genotype ST 10/CT 6742 (Appendix A). In this cluster, the contact between three animals from three different farms was traced back to the aforementioned mountain pasture in the district of Miesbach, mp-MB, in 2020. The respective isolates (15, 16, 20) were therefore assigned to the mp-MB outbreak (Appendix A). The genotype of this mountain pasture outbreak was found on 12 farms in two districts and the mp-GAP. Furthermore, two closely related isolates (Isolates 32 and 34) originated from two different farms in the district of Miesbach (Figure 4). The isolates were collected 8 months apart, and a transmission source could not be identified in the epidemiological analysis.

**Cluster 3** contained six strains from four different farms in the districts of Traunstein and Pfaffenhofen. These were exclusively isolated in the autumn and winter of 2020 (Table 1, Figure 3). Isolates 30 and 31, originating from the same farm, showed the same genotype ST 1487/CT 1324. Isolate 17 revealed an identical genotype and originated from a different farm in the district of Traunstein. A connection between these two farms could not be ascertained. In regard to another animal disease outbreak, two isolates (17 and 14) were collected from the same farm, but they revealed the different genotypes ST 1487/CT 1324 and ST 1487/CT 6738. These samples were taken 24 days apart (Appendix A).

Two more animal disease outbreaks on different farms in Cluster 3 revealed the same genotype ST 1487/CT 1324 (Isolates 26 and 27). One of these outbreaks (Isolate 27) emerged from a pig farm (Appendix A).

**Cluster 4** consisted of three isolates and three individual animal disease outbreaks in the districts of Bad Tölz-Wolfratshausen and Munich. The Cluster 4 genotypes exclusively revealed ST 1494/CT 8616. Each outbreak occurred within a time difference of 6 months (Appendix A). Epidemiological investigations revealed cattle trade between Farm 42 (Cluster 1, Isolate 6) and Farm 11 (Cluster 4, Isolate 8) in September 2019 (Appendix A).

**Cluster 5** also consisted of three isolates and two animal disease outbreaks solely in the district of Bad Tölz-Wolfratshausen. All isolates showed ST 1494/CT 8615. The epidemiological analysis revealed regular animal movements between Farm 16 (Isolate 9) and Farm 1 (Isolates 3 and 11) during the sampling period. Isolates 3 and 11 were collected at the same farm 6 months apart (Appendix A).

**Cluster 6** contained isolates, which caused outbreaks among cattle and occurred in the years 2017 and 2018. Both isolates revealed ST 10/CT 9033 (Appendix A). The epidemiological analysis determined a different district of origin and no contact between animals.

**Cluster 7** included strains from two different farms in the same district that were isolated 5 months apart (Appendix A). Again, no connection was discovered in the epidemiological analysis.

Three isolates (7, 10, 12) did not cluster with any other isolates and revealed three different genotypes, ST 1494/CT 8620, ST 1494/CT 8623, and ST 1487/CT 8626 (Table 1 and Appendix A).

Despite the different genotypes of Isolates 6 and 12, the epidemiological analysis revealed a joint transfer of animals originating from two farms in the district of Bad Tölz-Wolfratshausen (Table 1 and Appendix A; Figure 2 and Figure 4). The epidemiological analysis of all isolates except those originating from Cluster 1 revealed either a joint geographic focus or a temporal limitation (Table 1, Figure 2 and Figure 3).

## 4. Discussion

In the present study, *S.* Dublin isolates from bovines, pigs, and herding dogs in Southern Bavaria between 2017 and 2021 were characterized using antimicrobial susceptibility testing and whole-genome sequencing. Assembling these analyses and the epidemiological context, a comprehensive animal disease outbreak investigation was conducted.

### 4.1. Challenges Regarding Routine Laboratory Diagnostics and Animal Disease Outbreak Investigation

The sensitivity of detecting *S.* Dublin in clinical samples depends on the degree of pathogen colonization in the host animal. If clinically ill cattle shed concentrations of more than 100 colony-forming units per gram of feces, *S.* Dublin will be diagnosed with a probability of 60–100% even from rectal swabs [39]. According to prior publication, subclinically infected cattle shed lower amounts of bacteria, and the sensitivity rate drops to 16–20% [8]. In the present study, each index animal showed clinical signs of illness, but the majority of all investigated animals were apparently healthy. *Salmonella* hosted by intermittently shedding individuals might not be detected at all if samples are taken during a nonshedding period [11]. Hence, as mandated by German law, the number and timeline of required tests are specified prior to resetting the status of a formerly *Salmonella*-positive herd to negative [25]. In the present study, the corresponding investigations were conducted, which resulted in the diagnosis of 54 animal disease outbreaks. All listed outbreaks were resolved during the study time (Appendix A). Moreover, in the present study, genome sequencing was applied on selected outbreak isolates. High-resolution genotyping proved that most of the outbreaks revealed one individual genotype. As an exception, Isolates 2, 4, and 48 yielded an identical SNP-based genotype (Figure 5). Isolates 2 (cow origin) and 4 (dog origin) were collected from the same farm in July 2017. However, Isolate 48 was obtained from a cow belonging to a different farm and occurred 2 years later in August 2019 (Table 1 and Appendix A). Such a finding is unusual. Although the epidemiological analysis did not ascertain a connection, it is most likely that contact had taken place between animals for these two outbreaks. In this case, however, it remains unknown whether the source of the second outbreak was a still-carrying animal or an inanimate vector.

Various *Salmonella* serovars are known to potentially cause disease of variable severity in specific hosts [20]. This may be due to differences in pathogen virulence but also due to varying susceptibility of the host to a specific serovar [40]. Therefore, serotyping of *Salmonella* is crucial for human and animal diagnostics worldwide [20]. Furthermore, serotyping may support the delineation of an animal disease outbreak and reveal possible transmission routes [41]. In the present 5-year study, we found *S.* Dublin in two out of seven administrative regions in Bavaria (Figure 2). Despite the adaptation to cattle, three dogs also shed *S.* Dublin (Table 1). The pathogen could have caused an infection in the dogs, or the dogs could have been passive carriers. The 2 weeks of time between the sampling of Isolates 38 and 83 from the same dog could be indicative of an infection (Appendix A). Furthermore, classical serotyping is time-consuming and not always successful [42]. Our serotyping results were precisely reproduced by the application of the whole-genome “genoserotyping” tool (Appendix A). In regard to future whole-genome routine laboratory diagnostics, hands-on time may be reduced by switching directly to in silico genoserotyping. This procedure was previously been proven to be of value by other authors [29]. To date, genoserotyping still has its limitations in routine diagnostics of rare *Salmonella* serovars [43,44]. However, publicly accessible genome databases are growing rapidly so that the identification of rare *Salmonella* serovars and comprehensive genoserotyping will be possible in the near future [44].

### 4.2. Salmonella Dublin as a Zoonosis

As stated in a review comprising data for *S*. Dublin infections in humans from 1968 to 2013, the pathogen is currently considered an emerging pathogen [18]. Virulence factors contribute to the pathogenicity of *S*. Dublin isolates [45]. Furthermore, recent data revealed that the proportion of resistance in *S*. Dublin isolates in the United States is higher than in other *Salmonella* serotypes [18,19,46]. Furthermore, *S.* Dublin is one of the serotypes showing the highest multidrug resistance [19]. Data from the European Food Safety Authority (EFSA) indicate ongoing and overall high levels of antimicrobial resistance regarding *Salmonella* in the European Union [47]. However, the results of the present study revealed no significant antimicrobial resistance of the serovar Dublin via either in vitro or in silico approaches (Figure 1, Appendix A). The reliability of in silico antimicrobial susceptibility prediction of *Salmonella* isolated from bovine sources has been previously described, showing results similar to this study [48,49]. According to published worldwide comparison data, nearly all cases of antimicrobial resistance genes in *S.* Dublin seem to occur in the United States [50]. Previously published Germany-specific data support the results of our current study and reveal that multidrug resistance in *S.* Dublin is uncommon in this country [29]. The situation in Denmark, another European country, is similar to the German status of low multidrug resistance in *S.* Dublin [51]. Pursuant to German federal law, antibiotic treatment of livestock is restricted and closely monitored to curb antimicrobial resistance [52,53]. The treatment options list a serovar-specific *Salmonella* vaccine for the entire cattle herd as well as the elimination of carrier animals [54,55]. In comparison with the situation in North America, the repeatedly published low multidrug resistance and virulence of *S.* Dublin in Germany evidences a lower risk and better treatment situation for human patients with *S.* Dublin infection. This goal is in congruence with the One Health objective.

Concerning four previously reported human *S*. Dublin cases in Bavaria, three occurred in 2017 with no geographic proximity to animal cases. The fourth case, however, emanated from the district of Traunstein in 2020, the same district and year as Cluster 3 Isolates 14, 17, 26, 30, and 31 were collected (Figure 2, Appendix A) [24]. Salmonellosis in humans may emerge after their consumption of contaminated raw beef, raw milk, or raw milk cheese [18,56]. Patients with underlying chronic diseases are more susceptible to salmonellosis and severe disease progression [16,19,56]. In Denmark, an isolate collected from an infected human was linked to a beef isolate and the respective cattle herd [51]. The patient data of the Traunstein case were not obtained, and human isolates were not compared with isolates of the present study. Nevertheless, such a comparison would be an interesting project.

### 4.3. Epidemiology with a Special Focus on Mountain Pastures

*S.* Dublin has been repeatedly isolated in some regions of Germany, but the pathogen has never been described in certain other regions [29]. In the present study, all bovine isolates originated solely from one out of the seven administrative regions in Bavaria (Appendix A). From 2017 to 2021, a total of 152 outbreaks of bovine salmonellosis were detected in all of Bavaria. *S.* Dublin was the cause of 46% of these outbreaks (data not shown). The pasturing of cattle on alpine meadows is a unique tradition in Upper Bavaria [57]. In addition to animal welfare, the improved quality of the resultant food products distinguishes this favored tradition of cattle farming in the Alps [58]. In the summer months, cattle are herded communally on mountain pastures. Hence, this tradition poses some cross-infection risk for the cattle [12]. In the present study, mountain pastures were identified as a transmission focus for *S.* Dublin between animals of different farm origins. In assessing the broad variety of *S.* Dublin genotypes found in local connection to mountain pastures, we conclude that these represent a natural focus (Figure 4 and Figure 5). Similar observations have been made in Tyrol, the main district of alpine pasturing in Austria. The enzootic *Salmonella* infection of adult cattle was mentioned as a typical contagious disease [12]. Tyrol shares a border with Upper Bavaria.

According to prior observations, subclinical shedding of *Salmonella* has been linked to stress during transport [59,60]. At the beginning of the pasturing season in Bavaria, there are cattle drives, often over fairly long distances in unfamiliar terrain, to reach the mountain pastures [57]. Such transfer might cause stress in still carriers. An onset or increased shedding might then lead to *S*. Dublin accumulation on grazing grounds, thus posing an increased risk of infection for other animals.

As published, *S.* Dublin may last for at least 119 days in feces on grounds such as pastures in the summer and may also be viable after 87 days in surface water [9]. Wet pastures covered with fecal droppings that are contaminated with *Salmonella* provide ideal conditions for a fast spread of infection between different animals [12,61]. In the described outbreak on a mountain pasture in Garmisch-Partenkirchen, *S.* Dublin was isolated from 28 animals, including cattle and two dogs. Genotyping revealed that the isolates could be assigned to two clusters, three different cgMLST complex types, and nine branches of the phylogenetic SNP tree (Figure 5, Appendix A). This fairly broad variety of detected genotypes could be an indication that the outbreak may have lasted longer than the summer season of 2021 and might even exceed the present animal disease outbreak investigations. Our conclusion is supported by the fact that *Salmonella* may also survive in a viable but nonculturable condition in the natural focus of “alpine pasture” for an even longer time than the 119 days listed above [62].

In our study, we found isolates with the same cgMLST complex type over several successive quarters of years (Figure 3 and Figure 5). This finding is congruent with published data stating that *S.* Dublin may persist in cattle herds for years [8,9,51,63,64]. Isolates 3 and 11, 30 and 31, and 61 and 66 were isolated as pairs from the same farm, were assigned to the same outbreak, and revealed the same genotype. It was of note that the samples had been obtained up to 6 months apart (Appendix A). This particular phenomenon has been described previously for *Salmonella*, indicating a high degree of host adaptation and a concomitant low mutation rate at the nucleotide level. Consequently, this characteristic allows epidemiological investigations using whole-genome analysis [26,65,66].

Due to the high prevalence of *S.* Dublin in Upper Bavaria, we are referring to this administrative region as “endemic” and list the prevalent ST and CT genotypes in this study (Table 1 and Appendix A). Our data show that in an endemic region, animal disease outbreaks can be caused by multiple genotypes. The high-resolution outbreak investigation of the mp-GAP revealed three different genotypes (Table 1). Thus, whole-genome sequencing of multiple isolates per outbreak provides valuable information for epidemiological investigations. Sequencing one isolate per outbreak or using serotyping as the only method may not provide sufficient information. The data in this study could not confirm previous epidemiological investigations that distinguished isolates from endemic regions from those from nonendemic regions [26,29]. However, a thorough traceback to one single index case per outbreak was not successful in this study. Again, this limitation has been described previously [26]. Since *S.* Dublin may persist in herds or natural foci, the initial epidemiological contact could have occurred years ago. As published, the distribution of *S.* Dublin in cattle has been consistently linked to transport or movement of infected animals [65].

As a “lessons learned” conclusion, cattle should be screened for salmonellosis before being returned to their farms in the fall and before the pasturing season in the summer [12]. Moreover, internal biosecurity measures are recommended for *S.* Dublin control in herds within an endemic region [51]. In addition to culling *S.* Dublin-positive tested cattle, cleansing and disinfection of stables before housing cattle again has proven effective [67]. On alpine pastures, however, biosecurity measures prove to be problematic since there are hardly any boundaries for cattle and wildlife. *S.* Dublin is considered host adapted, and control measures are therefore focused on cattle [8]. The detection of *S.* Dublin-positive dogs and pigs in our study and multiple other species in previous studies [9,68,69] may be an indication that the focus should be expanded. Further investigations are needed to clarify whether other species represent reservoirs or a source of infection for cattle.

Next-generation sequencing technology provides valuable information. A single testing method provides information about phylogenetics, antimicrobial resistance, and virulence [44]. Regarding One Health, next-generation sequencing enables a rapid and focused response in *S.* Dublin outbreaks.

## Figures and Tables

**Figure 1 microorganisms-10-00885-f001:**
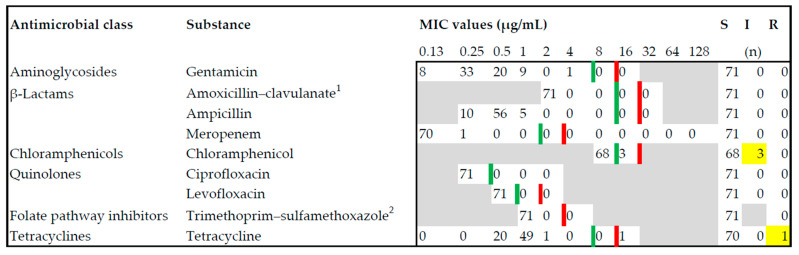
Antimicrobial susceptibility testing for *Salmonella* Dublin isolated from bovines, pigs, and dogs was carried out on nine substances from six antimicrobial classes. Red lines delineate the breakpoints towards resistant (R), and green lines delineate the breakpoints towards intermediate (I) minimum inhibitory concentration (MIC). Yellow boxes indicate the only intermediate and resistant isolates found in this study. Most of the isolates were completely susceptible (S). ^1^ Concentration of amoxicillin is reflected, concentration ratio of amoxicillin–clavulanate: 2:1. ^2^ Concentration of trimethoprim is reflected, concentration ratio of trimethoprim–sulfamethoxazole: 1:19.

**Figure 2 microorganisms-10-00885-f002:**
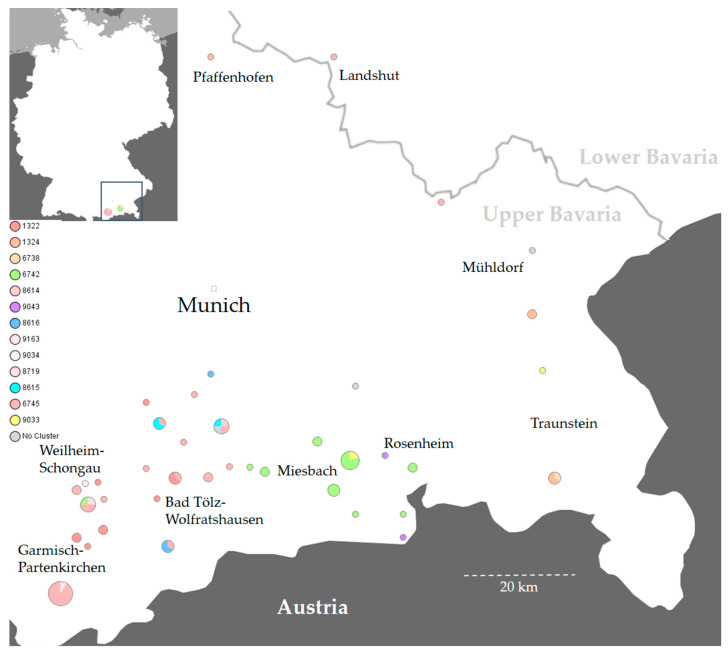
A map of Germany (top left) and the respective enlarged map extract depict the two administrative regions of Upper and Lower Bavaria, the state capital Munich, and various districts in Southern Bavaria affected by *Salmonella* Dublin in the 2017–2021 period. The individual clusters and cgMLST complex types (CTs) are highlighted in different colors. A CT legend is reflected on the left side of the map. The circle size represents the number of identified isolates. Cluster 1 (red shades) is dispersed throughout Southern Bavaria, whereas other clusters have a local focus. Isolates that could not be assigned to any cluster are marked in gray. The map was created with SeqSphere+ and Microsoft PowerPoint.

**Figure 3 microorganisms-10-00885-f003:**
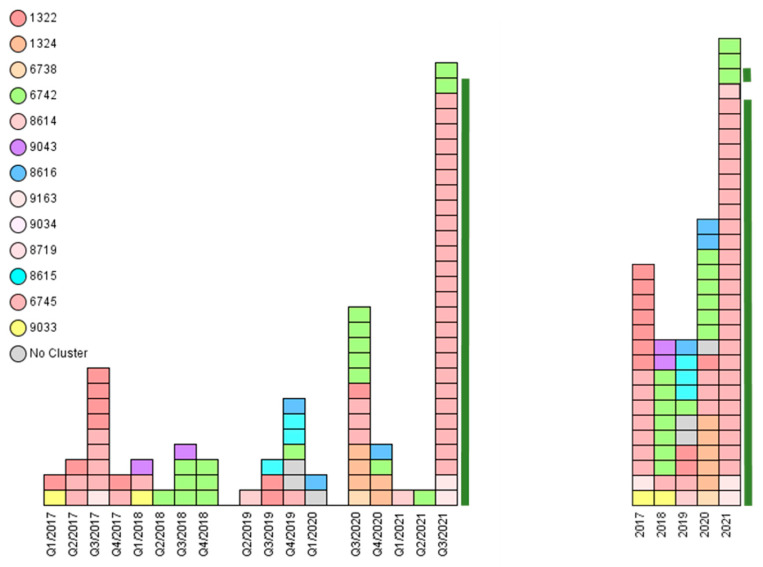
The epidemiological analysis of *Salmonella* Dublin animal disease outbreaks in Southern Bavaria revealed assorted genotypes within the 5-year investigation period. Each rectangle represents one isolate. Left: Complex types (CTs) distributed over the quarters (Q1 to Q4) of the years 2017 to 2021. Right: CTs distributed over the years 2017–2021. Most CTs persist over the years. Cluster 1 prevailed in 2017 and 2021. The 28 isolates from a 2021 outbreak on a mountain pasture in Garmisch-Partenkirchen are depicted with a green bar.

**Figure 4 microorganisms-10-00885-f004:**
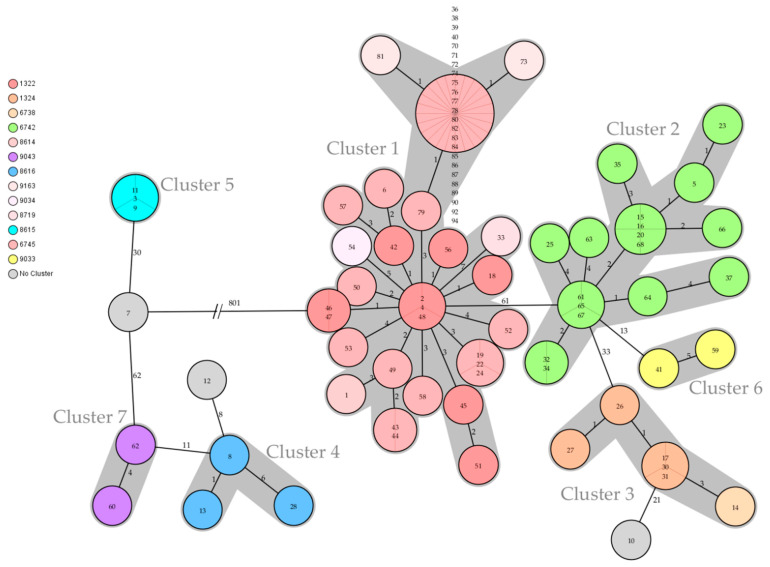
The minimum spanning tree shows all seven clusters (gray shades) of *Salmonella* Dublin found in Southern Bavaria in the 2017–2021 period. Three isolates could not be assigned to any cluster (gray circle without shade). The colored circles visualize the various cgMLST complex types with the corresponding four-digit numbers listed on the left side of the tree. The circled numbers represent the respective isolate numbers, and the circle size corresponds to the number of isolates sharing the same genotype. The numbers on the connecting lines represent the allele difference between two circles. The tree is based on cgMLST values and SeqSphere+ computations.

**Figure 5 microorganisms-10-00885-f005:**
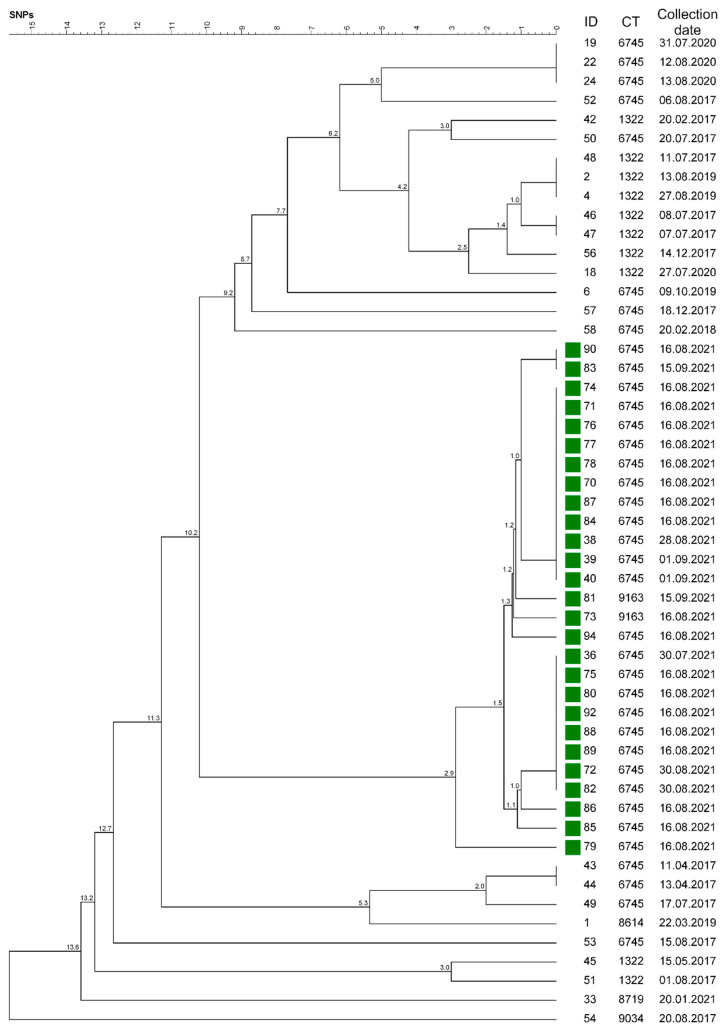
The phylogenetic tree was calculated on the basis of single-nucleotide polymorphisms (SNPs), including all *Salmonella* Dublin isolates (ID) that revealed a Cluster 1 genotype. The corresponding four-digit cgMLST complex types (CTs) and the collection date of the respective isolate are listed on the right. The isolates marked in green originated from the mountain pasture outbreak in Garmisch-Partenkirchen and revealed a minimal distance of 2.9 SNPs to the latest ancestor. The tree was generated using BioNumerics.

**Table 1 microorganisms-10-00885-t001:** Metadata and cgMLST results for 88 *S.* Dublin isolates (2017–2021) from 54 animal disease outbreaks were investigated in this study (compare Appendix A).

Cluster	Complex Type	Isolate Number	Year	Host	Number of Farms	District
1	1322	42, 51, 56	2017	cattle	3	TÖL
1	1322	45	2017	cattle	1	WM
1	1322	46, 47, 48	2017	cattle	3	GAP
1	1322	2, 4	2019	cattle, dog	1	GAP
1	1322	18	2020	cattle	1	TÖL
1	6745	43, 49, 50, 53	2017	cattle	4	TÖL
1	6745	44	2017	cattle	1	MÜ
1	6745	57	2017	cattle	1	WM
1	6745	52	2017	cattle	1, mp-GAP	GAP
1	6745	58	2018	cattle	1	TÖL
1	6745	6	2019	cattle	1	TÖL
1	6745	19, 24	2020	cattle	2	TÖL
1	6745	22	2020	pig	1	LA
1	6745	36	2021	cattle	1	TÖL
1	6745	39, 40, 70–72, 74–78, 80, 82, 84, 85–90, 92	2021	cattle	6, mp-GAP	GAP
1	6745	38, 83, 94	2021	dog	mp-GAP	GAP
1	6745	79	2021	cattle	mp-GAP	GAP
1	8614	1	2019	cattle	1	TÖL
1	8719	33	2021	cattle	1	GAP
1	9034	54	2017	cattle	1	WM
1	9163	73, 81	2021	cattle	2, mp-GAP	GAP
2	6742	61, 66	2018	bovine	1	RO
2	6742	63, 64, 65, 67, 68	2018	cattle	5	MB
2	6742	5	2019	cattle	1	RO
2	6742	15 *, 16 *, 23	2020	cattle	3, mp-MB *	RO
2	6742	20 *, 25, 32	2020	cattle	3, mp-MB *	MB
2	6742	34, 35	2021	cattle	2	MB
2	6742	37	2021	cattle	1, mp-GAP	GAP
3	6738, 1324	14, 17	2020	cattle	1	TS
3	1324	26, 30, 31	2020	cattle	2	TS
3	1324	27	2020	pig	1	PAF
4	8616	8	2019	cattle	1	TÖL
4	8616	13	2020	cattle	1	M
4	8616	28	2020	bovine	1	TÖL
5	8615	3, 9, 11	2019	cattle	2	TÖL
6	9033	41	2017	cattle	1	RO
6	9033	59	2018	cattle	1	TS
7	9043	60, 62	2018	cattle	2	RO
none	8620	7	2019	cattle	1	RO
none	8623	12	2019	cattle	1	TÖL
none	8626	10	2019	cattle	1	MÜ

mp: mountain pasture, GAP: Garmisch-Partenkirchen, MB: Miesbach, TÖL: Bad Tölz-Wolfratshausen, WM: Weilheim-Schongau, MÜ: Mühldorf, LA: Landshut, RO: Rosenheim, TS: Traunstein, PAF: Pfaffenhofen, M: München. * is indicating isolates from “mp-MB *”.

## Data Availability

The raw sequencing data were deposited at the European Nucleotide Archive, ENA project number: PRJEB50766.

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
