# Peer review of "Whole-Genome Investigation of Salmonella Dublin Considering Mountain Pastures as Reservoirs in Southern Bavaria, Germany"

_microorganisms, 2022, doi:10.3390/microorganisms10050885_

Round 1

Reviewer 1 Report

It is a good paper working on Salmonella Dublin about Epidemiological Survey.I have some suggests:

1,The title, it is easy to be misunderstood, "Whole Genome Investigation of Salmonella Dublin Outbreaks in Southern Bavaria, Germany" or the other maybe better

2,line 28, the lastest reference shall be used,   "more than 2600 serovars"

3,I do not  find the content in the methods about virulence and plasmid in the results, please revise it 

4,italic special name in references

Author Response

Please see our response in the attached pdf.

Thank you!

Reviewer 2 Report

Manuscript very difficult to read because the English is to be reviewed and the information is partial, poorly expressed and poorly organized. It is a tribute because the study can be interesting. It could have been interesting to recover enterobase genomes for cgMLST analysis. I don't know if there are any from this region of Germany but it would be interesting to recover all the genomes of bovine strains isolated in Germany and include them in the cgMLST analysis to see if the genotypes identified in Bavaria are really specific to this region and these pastures. Accessory genome data are not used or discussed. The major results on the accessory genome should be inserted into a heat map associated with the SNP tree analysis for more clarity. SNP distancies should be calculated for all the profiles identified and standard deviation calculated.

Author Response

Please see our response in the 2 attached pdfs.

Thank you!

Reviewer 3 Report

The authors present a study on “ Whole Genome Outbreak Investigation of Salmonella Dublin including Mountain Pastures in Southern Bavaria, Germany”. In my opinion, there are some minor issues present in the manuscript that must be resolved. The authors should carefully address all of the raised issues that are explained in detail below and must apply the changes in the manuscript accordingly.

Specific comments:

Introduction

  • Lines 32-33: The sentence “ Strong characteristics of a Salmonella infection and frequently described for the serovar Dublin are still carrying and subclinical shedding” is not clear. Please rephrase it.
  • Lines 60-63: In my opinion in the introduction is important to state the aim. Other detailed information (Number of outbreaks and number of Salmonella isolates) should be included only in the Materials and methods section

Materials & Methods

  • Line 82: Don’t you know the total number of Salmonella outbreaks (2017-2021)? If yes, please indicate it.
  • Lines 86-87: The sentence “ In regard to other questions such as mixed animal species on farms or deferred sampling, additional isolates were included” is not clear
  • Lines 95-104: In my opinion, All the information reported here should be not included in Materials and Methods but either in the introduction or in the discussion
  • Line 119: Please report the number of Figure, in a bracket, in which the antimicrobial used are reported (Figure 1).

Discussion:

  • Line 402: Please change the name of the paragraph.

Author Response

(The authors gave the same response as above.)

Reviewer 4 Report

The article is very interesting, but requires some corrections, I am enclosing my suggestions. 

The article: Whole Genome Outbreak Investigation of Salmonella Dublin including Mountain Pastures in Southern Bavaria, Germany

The topic covered in this article is very interesting and could be developed in the future.

The introduction is written very clearly and closely related to the subject of the article.

The Materials and Methods describe the techniques used to isolate Salmonella spp and their sources. The methods of DNA isolation, sequencing and analysis of the results were presented, as well as the relevant validation reference.

Line 12, line 74 and line 180: “and two in bovine herds other than cattle”

“bovine other than cattle”- could you explain what the authors meant. Do the authors know what type of herd these samples came from?

I have a few comments to present the results:

“In the present study, 54 S. Dublin outbreaks were analysed over a five-year period by using phenotypic and whole genome characterization on 88 isolates to establish antimicrobial resistance, virulence potential and phylogenetic analysis in order to perform trace back outbreak investigations of S. Dublin in Southern Bavaria”

Line 189-198 and Figure 1

The 88 tested strains cannot be counted, the text shows that 74 strains and "all others" and in the table (fig. 1) the tested strains do not add up to 88 in any way. In my opinion, the tables should be clear and present all the results, even those negative.

The description for Fig1 is also too modest - there is no source of origin (animals) of the studied strains.

Often the result tables in on-line publications are viewed in a new window, and they should clearly correspond to "samples in M&M"
Line 218 The title of Table 1 is unclear and it is not known what is shown in it

sample number 52 is presented twice in Table 1:
line 9 of table 1.: 1 6745 52 2017 cattle 1, mp-GAP GAP
line 15 of table1: 1 6745 39, 40, 52, 70-72, 74- 78, 80, 82, 84, 85-90, 92 2021 cattle 6, mp-GAP GAP

Line 248 – 251 One dog was sampled twice which revealed genetically identical isolates (38 and 83), whereby the latter was collected four weeks later. Duplicate samples from two cows taken two weeks apart also revealed an identical genotype (39 & 71 and 75 & 82). 

Samples should be described in detail in M&M samples, especially if they do not differ and are collected in close time.

Author Response

(The authors gave the same response as above.)

Round 2

Reviewer 2 Report

The article was proofread by a native English speaker but there was no effort to reorganize the results to make the message of the study clearer. The drafting of the discussion has not been reworked either and the request to review the geographical map has not been taken into account.

As I suggested earlier it would be interesting to analyze the 88 genomes with these present in Enterobase (there are 94 Salmonella Dublin isolated in Germany). Take into account, for example, hypotheses related to wild animals or waterways in the region.

My other suggestions in the pdf.

Author Response

Response to Reviewer 2 Comments

Point 1: The article was proofread by a native English speaker but there was no effort to reorganize the results to make the message of the study clearer. The drafting of the discussion has not been reworked either and the request to review the geographical map has not been taken into account.

Response 1: Right, we reorganized the results according to several suggestions. We also reworked the disccussion section. The map was revised. Please find the changes in the submitted manuscript.

Point 2: As I suggested earlier it would be interesting to analyze the 88 genomes with these present in Enterobase (there are 94 Salmonella Dublin isolated in Germany). Take into account, for example, hypotheses related to wild animals or waterways in the region.

My other suggestions in the pdf.

Response 2: Our main interests lie in routine diagnotics and our work included 88 isolates collected in five years. We compared these among each other and the raw sequencing data were deposited at the European Nucleotide Archive, ENA project number: PRJEB50766. In our opinion our study is comprehensive regarding the questions of our main interests. If further theories are to be addressed we are open to provide the isolates and to collaborate in a future project.

We considered the comments of the pdf in our revised manuscript and also replied to it in the included revised pdf version.

-   please see as well the attached pdf document - 
